# A Network Analysis of Research Topics and Trends in End-of-Life Care and Nursing

**DOI:** 10.3390/ijerph18010313

**Published:** 2021-01-04

**Authors:** Kisook Kim, Seung Gyeong Jang, Ki-Seong Lee

**Affiliations:** 1Department of Nursing, Chung-Ang University, Seoul 06974, Korea; kiskim@cau.ac.kr; 2College of Nursing, The Catholic University of Korea, Seoul 06649, Korea; 3Da Vinci College of General Education, Chung-Ang University, Seoul 06974, Korea; goory@cau.ac.kr

**Keywords:** text mining, semantics, nursing methodology research, terminal care

## Abstract

This study identified the trends in end-of-life care and nursing through text network analysis. About 18,935 articles published until September 2019 were selected through searches on PubMed, Embase, Cochrane, Web of Science, and Cumulative Index to Nursing and Allied Health Literature. For topic modeling, Latent Dirichlet Allocation (K = 8) was applied. Most of the top ranked topic words for the degree and betweenness centralities were consistent with the top 1% through the semantic network diagram. Among the important keywords examined every five years, “care” was unrivaled. When analyzing the two- and three-word combinations, there were many themes representing places, roles, and actions. As a result of performing topic modeling, eight topics were derived as ethical issues of decision-making for treatment withdrawal, symptom management to improve the quality of life, development of end-of-life knowledge education programs, life-sustaining care plan for elderly patients, home-based hospice, communication experience, patient symptom investigation, and an analysis of considering patient preferences. This study is meaningful as it analyzed a large amount of existing literature and considered the main trends of end-of-life care and nursing research based on the core subject control and semantic structure.

## 1. Introduction

End-of-life (EOL) care is a global concern. It refers to providing support to people in the last months or years of their lives [1]. The importance of providing quality EOL care is recognized worldwide [2]. Providing timely EOL care is challenging as it is difficult to predict when residents are nearing death because frail older adults may die without a clear terminal stage and the trajectories of dying also vary [3]. During EOL care, many patients experience life events, where they have insufficient control over important changes in their lives leading to confusion [4]. These uncertainties may lead to conflict among residents, family members, and caregivers. Nurses are well-versed in the physical and psychological conditions and preferences of their patients and hence, must cope with these complex needs [5].

EOL care is an essential provision in long-term care facilities with multidisciplinary teams, including nurses, assistants, physicians, social workers, and volunteers. All the members require a clear description of their role for an effective collaboration [5,6]. Critical care nurses play an integral role in supporting older patients and their families facing the EOL care decision-making challenge. Despite national imperatives to improve the quality of EOL care, patients continue to experience uncontrolled pain, inadequate communication, and life-sustaining treatment against their will. These contribute to increased hospitalization periods and costs. Nurses must understand the prevalent issues to address the needs of such patients [7].

International research and policy development have been attempting to identify the common features of EOL care and encourage innovative ways to provide effective and culturally appropriate approaches [2,8].

Nevertheless, research to develop an efficient EOL care program based on a comprehensive approach to understanding EOL care is limited. Existing methods for a thorough analysis are limited in time, labor, and accuracy, when working with extensive research topics and the large amount of literature. The text network analysis method is used in various disciplines, including nursing, because it is useful for analyzing a smaller number of subjects within a large amount of text data as well as “big data” using a computer program for social network analysis [9,10,11]. This analytical technique could add new insights into the nursing research field with methodological flexibility and epistemological diversity; however, few studies employing this technique have been conducted in the nursing field [12]. Furthermore, data analysis techniques, such as network analysis that describes group dynamics, can be used to understand how multiple individuals coordinate, communicate, and make decisions about EOL management [13].

Therefore, this study identified the objective trends that could be confirmed by analyzing the end-of-life care and nursing literature. The text network analysis was used to examine how the research has been conducted in terms of topics, methods, and approaches.

## 2. Materials and Methods

### 2.1. Study Design and Research Procedure

In this study, a text network analysis was performed to search for important keywords and research topics by building a network based on the co-occurrence of the titles of EOL care and nursing research literature and the keywords extracted from the abstracts.

Titles and abstracts from published EOL care studies constituted the data range. The relationship between concurrent keyword emergences was identified to investigate EOL care research trends and characteristics of the nursing field.

### 2.2. Process of Search and Selection

Researchers collected relevant literature on EOL care and nursing through searches on PubMed, Embase, Cochrane, WOS (Web of Science), and CINAHL (Cumulative Index to Nursing and Allied Health Literature). A search for articles containing the keywords “end-of-life care” and “nursing” in the title or abstract revealed 18,935 articles published until September 2019 (Table 1). About 7419 articles were selected after excluding duplicate ones (Figure 1). Duplicate publications were managed with the EndNote program. The titles and abstracts of the selected articles were organized using Microsoft Excel.

### 2.3. Keyword Extraction and Preprocessing of the Titles

Before the analysis, the Natural Language Toolkit (NLTK) was used for word preprocessing. First, all sentence codes and numbers from the article titles and abstracts were removed. Second, we extracted all the words by separating them with spaces and refining them to their unique form, based on lemmatizing to facilitate plentiful text analysis. Lastly, words that have no specific and direct meaning, such as “this,” “what,” “the,” and “an,” and numbers or symbols were set as “stop words” and removed. Thus, through this process, a corpus (word set) was created.

### 2.4. Keyword Analysis

Words that appeared together in the article titles were defined as “related.” After extracting all pairs of words that coexisted in the article titles, duplicate combinations were removed. A pair represents an edge, and each edge contains information about the start and end nodes of the connection. The word network we created contained 5512 nodes and 130,667 edges. Each node in a given keyword network has one or more “edges,” and the total edges of a given node constitute the “degree.” Degree centrality refers to a technique that measures the number of adjacent nodes and determines how the node plays a central role in the graph. Betweenness centrality determines how a node plays a central role in its connection with other nodes. Any two nodes have at least one shortest pathway that connects them, and betweenness centrality measures how many shortest paths pass through a node when there is only one. The degree centrality value of each node is expressed by normalizing the degree from 0 to 1 by dividing the node’s degree by the maximum degree value in the graph. Betweenness centrality is also normalized to a value between 0 and 1 divided by the total number of edges a graph may have [14].

### 2.5. Topic Analysis

Topic modeling is an analysis method that can identify the topics hidden in a large amount of text data. In this study, the most commonly used Latent Dirichlet Allocation (LDA) method was applied [15]. LDA is a probabilistic method of inferring topics by assuming that a document is composed of several topics, which are a set of keywords [16,17]. LDA finds topics that are commonly covered in several documents. For example, if words such as “dog,” “puppy,” “cute,” “snack,” and “walk” frequently appear together in multiple documents, LDA collects these words and treats them as a topic. A topic is a set of words that are automatically calculated through computation. LDA allocates words by iteratively executing an algorithm to find the best topic. A topic is a set of words and not a name. Therefore, topic analysts need to see [dog, puppy, cute, snack, walk] and name them, such as “pet dog;” therefore, expertise for data interpretation is required. Additionally, “animal hospital” topics such as [dog, sick, hospital, emergency] may additionally be discovered. In this case, it is possible that one word, such as “dog,” appears in multiple topics.

In the LDA technique, the researcher can determine the optimal number of topics that are best categorized by calculating the number of topics (k). Additionally, we used clustering results obtained from keywords of a specific topic to set the topic’s k-value. Based on clustering, the number of topics in the included abstracts was assumed to be 10. Subsequently, we reviewed and discussed the topic modeling results. Topics were labeled with a name that could collectively express the keywords included in each topic. This process was implemented in the Python environment using MALLET (http://mallet.cs.umass.edu/), which is publicly accessible for topic modeling.

## 3. Results

### 3.1. Core Keywords that Emerged from the EOL Care Studies

Keyword ranking was based on the frequency of the appearance of keywords used in the EOL studies’ titles (i.e., the number of articles with keywords, degree centrality, and betweenness centrality). Table 2 lists the top 30 core keywords by criteria. The frequency order of the simple appearance was as follows: “care” (5655), “end-of-life” (2973), “patient” (1503), “palliative” (1380), and “nurse” (1329). Identical results were observed in the top five degree and betweenness centralities.

The top five ranked words for the degree and betweenness centralities were as follows: “care” (0.6991/0.2583), “end-of-life” (0.5304/0.1253), “patient” (0.3948/0.0620), “palliative” (0.3647/0.0450), and “nurse” (0.3391/0.0404). Degree centrality is to find out which keywords have a lot of connection to the research topic. The words that are ranked at the top are the ones that are being treated in focus. On the other hand, it can be seen that research is insufficient for words such as “dementia” (rank 25th), “qualitative” (rank 26th), and “old” (rank 28th) that are ranked in the lower part. Betweenness centrality is a measure of the degree to which one keyword acts as an intermediary to another. Although degree centrality and betweenness centrality show similar patterns, there are keywords that show differences. Keywords with high betweenness centrality can have a great influence on controlling the flow of information within the network. The word “support,” ranked 28th in betweenness centrality, does not exist in the degree centrality ranking. This means that “support” has a greater role in affecting other words than its central role in words.

The frequency, and degree and betweenness centralities for the top eight words were similar. However, in terms of frequency and degree centrality, the word “home” ranked 6th and 7th, respectively, and its betweenness centrality ranked lower at 10th. The word “home” is more connected to the other frequently used words seen as high in importance. However, it can be interpreted that centrality was relatively low among other words. The word “dementia” ranked 15th for frequency in the titles, and both degree and betweenness centralities were lower at the 25th rank. Thus, although, “dementia” is often mentioned, its importance and centrality are relatively low. This difference in analysis shows that the frequency of the keyword is not necessarily proportional to the size of centrality. In other words, it means that even keywords with high frequency may not be relatively centered.

Compared to “family” and “people,” “patient” appeared more frequently in the EOL care studies and showed a higher connection and centrality, thus indicating that more studies were conducted on patients in hospitals rather than on general and healthy population. Furthermore, words with similar forms such as “nurse” and “nursing,” “die” and “death,” and “quality” and “qualitative” were close to each other and had similar rankings in frequency and centrality. Since words of the same meaning with different forms are considered to have similar positioning, priority results are reliable.

### 3.2. Semantic Network Analysis

Figure 2 displays a semantic network diagram that was created to examine the network between core keywords based on degree centrality. These are the nodes ranked in the top 1% with a degree of more than 600. The words in the Figure 2 have more than 600 connecting edges. Higher-ranked nodes are located in the center, and lower-ranked nodes are located in the periphery. Through the entire semantic network, it can be seen that words are grouped around “care.” Nodes in the center have a greater influence than nodes located in the periphery. It has a similar ranking to the words located in Table 2.

The keyword with the highest degree centrality was “care.” The keywords ranked in the top 1% of degree centrality were “end-of-life,” “patient,” and “palliative,” which were consistent with the most frequently observed keywords.

### 3.3. Keywords by Five-Years Cycle

Keywords that appeared until 2019 were listed in a five-year cycle (Figure 3). Words are listed according to the number of frequencies by five-year cycle, and the number of frequencies is written in parentheses. Words that rise above three levels are connected by a red line, words descending more than three levels are connected by a blue line. If there is a change in less than three levels, it is marked with a gray line. Additionally, there is no connection line when the word is not in the top 30. “Care” was unrivaled as the most frequently used keyword in all periods. “Patient” and “nurse” were in the top five. “Palliative” began to be used from the late 90s and reached from the lowest in the rank to the top four in the most recent cycle. “End-of-life” was rarely used in the early 1990s; however, its frequency gradually increased from 4 (1.2%; between 1991 and 1995) to 67 (7.6%; between 1996 and 2000) and 1038 (11.4%; between 2016 and 2019). “Terminal” is from 29 (3.3%; between 1996 and 2000) to 27 (1.2%; 2001–2005), “terminally” is from 17 (1.9%; between 1996 and 2000) to 29 (1.3%; 2001 and 2005), dramatically decreased from the 2000s and disappeared from the upper frequency.

There were also words that rise markedly at every point in time. “Home,” “death,” “support” in 1990, “end-of-life” in 1995, “palliative” in 2000, “hospice,” “study” in 2005, “experience” in 2010 “dementia,” “quality,” “advance,” and “review” in 2015. On the other hand, there are “hospital,” “terminally” in 1995, “terminal,” “terminally” in 2000, “hospice,” “quality” in 2005, “practice,” “intensive,” and “unit” in 2010. As the clear definitions of “end-of-life” and “hospice” emerged, the use of ambiguous words such as “terminally ill” and “terminal stage” seems to have declined.

### 3.4. Frequency According to Contiguous Sequence Word Analysis of Research Titles

Table 3 shows the results of contiguous sequence word analysis. Owing to low frequencies, the cases of four- and five-word expressions of contiguous words were excluded; the two- and three-word processing for the top 20 were shown as follows. Contiguous sequence is not a simple frequency of two- and three-word phases. This section shows not only the co-occurrence of words appearing together based on the centrality between words, but also in what order they appear by focusing on the contiguous sequence. This represents a directed path between words.

“End-of-life + care” was the most frequent (1829), followed by “palliative + care” (1071) and “nursing + home” (390). According to the results of the three-word analysis, “intensive + care + unit” was the most frequent (255 cases), followed by “advance + care + plan” (197 cases) and “nursing + home + resident” (127 cases).

When “unit” and “home” were placed after the word, the word clusters referred to a place: “nursing + home,” “care + unit,” “care + home,” “intensive + care + unit,” and “care + nursing + home.” When “nurse” was added, the words meant the role of a nurse: “care + nurse” and “critical + care + nurse.” The words “care + plan” and “advance + care + plan” were related to documents for palliative care and were ranked at the top. Particularly, “care” was most frequently used in both two- and three-word expressions. “Care” was a common term included in words that represented a place, activity, or role. When “care” was added to the end of a contiguous sequence, it represented activities such as “end-of-life + care,” “palliative + care,” “intensive + care,” “critical + care,” “long-term + care,” and “palliative + end-of-life + care.” When “patient” and “people” are placed after the words, they represent subjects for nursing performance: “cancer + patient,” “care + patient,” “die + patient,” “care + die + patient,” “ill + cancer + patient,” “terminally + ill + patient,” “end-of-life + care + patient,” “end-of-life + care + people,” and “palliative + care + patient”.

### 3.5. Topic Analysis of the Abstract

The collected texts were analyzed using the LDA algorithm. LDA topic analysis identifies topics commonly contained in literature based on unsupervised learning. The topics are formed as keywords combination based on statistics. Experts in the relevant field judge the meaning of keyword combinations according to statistics and derive meaningful topics. Seven rounds of LDA were performed with varying number of topics (K = 2, 3, 4, 5, 8, 10, 20). In the case of K = 2 and 3, it was difficult to derive meaningful content because the number of subtopics was too small. Since K = 20 has a large number of subtopics, there were overlapping topics. Subtopics were grouped through discussion among researchers, and K = 8 topics with no overlapping meaning between groups were finally selected (Table 4). Each topic was ranked with reference to word weight, and the top 20 collocates in the corresponding topic were extracted. Weight is a value that represents the weight of each topic in the texts collected; it was expressed in a range of 0–1, where 1 indicated the most weighted. Meaningful keywords were combined to group each subtopic. Eight subtopics that can contain keywords were named. This process is similar to content analysis.

Ethical problems of the decision to end the treatment: there are “decision,” “ethical,” “euthanasia,” “sedation,” and “withdrawal” in order of weight. Additionally, words such as “decision-making,” “suicide,” and “withhold” are topics related to patient self-determination, such as discontinuation of treatment. The subtopic of these words was interpreted as a study relating to ethical problems based on what the patient’s earlier statements, euthanasia, and suicide. There are still no clear answers to the problems associated with withdrawal or withholding life-sustaining treatment. Studies on ethical issues arising between the patient’s right to self-determination and the obligations of medical personnel are being conducted.Symptom management to improve the quality of life: among the top weight words, the word “cancer” was differentiated from other groups. There were also “symptom,” “quality,” “pain,” and “life”. Based on this, these are focused on pain interventions to improve the quality of life of cancer patients. Pain management is one of the important factors in EOL care. Especially, pain management for cancer patients is an important influence on the patient’s quality of life.Development of EOL knowledge education programs: this subtopic group contains words for education such as “student,” “practice,” “program,” and “knowledge.” Additionally, not only “student” but also “nurse” and “staff” were included, so the subject of education was not limited. It was interpreted as a study developing and researching education programs pertaining to life-sustaining treatment and palliative care for nurses and nursing students. With an increase in the demand for EOL care, the need for education programs for medical personnel is increasing. Consequently, it was confirmed that program development studies were in progress.Advanced care planning for older adults: “advance,” “preference,” “plan,” and “decision” are terms related to “ACP (advanced care plan)” and “AD” (advance directives). In particular, the subjects were limited to the elderly through the words “dementia” and “old.” This refers to writing an advanced directive and advanced care planning upon discussion with a physician (or resident) to reflect adult patients’ preferences and wishes. There are ethical problems related to older and dementia patients. Currently, many studies are being conducted on the life-sustaining treatment of elderly patients. There have been studies to focus on advance planning of care treatment in case of chronic diseases such as dementia.Home-based hospice: this subtopic shows the phrases “home,” “people,” “death,” and “place,” which is the place that people prefer. It could also be deduced from the words “community,” “support” that there are studies of home-based hospice in the community. Home, hospital, and community need to provide a place for care. In addition to hospital-based hospice services, home-based hospice services are provided. Studies on how to provide such services were conducted.Communication experiences: it was found that “patient,” “nurse,” and “family,” which refer to the subjects, took the top place, and the words “experience,” “interview,” and “communication” describe the communication with the subjects. These include interview studies to survey EOL communication experiences in patients, caregivers, and healthcare providers. It is important to understand the patient’s preferences in EOL treatment. Accordingly, studies on communication experiences and methods were performed.Survey of patient symptoms: there are words for the subject such as “patient,” “family,” “caregiver.” Additionally, words “symptom,” “pain,” “quality,” “questionnaire,” “score,” “survey,” “measure,” and “scale” imply the meaning of examining the subject’s symptoms such as pain and quality. Methods of surveying pain and symptoms in EOL patients include scoring, scales, and questionnaires. It may appear to be the same as “2. Symptom management to improve the quality of life,” but the focus here was on the research methods and tools to be used to investigate the symptoms.Analysis of considering patients’ preferences: after the high weight word “hospice,” there are the words “home,” “hospital,” “facility,” “day,” “year,” and “receive”. This was interpreted as representing studies in which patients conducted surveys on preferred (“likely”) places and schedules related to hospice or palliative care and confirmed CI (confidence interval). There are studies in which physicians provided EOL care after receiving information about the desired place (home, hospital, or institution) and date (day and year) of care from the patient. Studies have been conducted on whether checking and applying the patient’s preferences yields better results.

## 4. Discussion

This study collected literature on EOL care in the field of nursing published until 2019. We performed a text network analysis to identify the major study topics pertaining to EOL care in nursing and elucidated the specific discipline in which the highest number of research topics was discovered. After extracting frequently appearing keywords, a network was established to analyze the degree of connection and centrality. Subsequently, research keywords pertaining to EOL care and nursing were classified by topic using the LDA method to examine research trends. The core keywords and features of major meaning structures observed in the network analysis are discussed below.

First, the top five keywords in terms of frequency, degree centrality, and betweenness centrality in EOL care and nursing-related literature were “care,” “end-of-life,” “patient,” “palliative,” and “nurse.” These were consistent with the keywords ranked in the top 1% in the semantic network analysis. Internationally, “hospice palliative care” and “end-of-life care” are often used as set expressions. Although, the definition of the terms varies across countries or institutions [18], the meaning is identical. Furthermore, combination of frequently used terms showed that previous studies had examined the place of EOL care, the nurses’ roles, and relevant documents. The most frequent word “care” has many diverse aspects in nursing—spiritual care [19], tender care [20], and personal care [21] for life sustaining patients, but these were not included in the top results extracted during analyses.

Second, we analyzed the yearly trends in research topics to determine topics in trend. “End-of-life” was rarely used in the early 1990s, but its use increased considerably in recent years, while the use of “terminally ill” declined. A similar distribution of terms was observed in a systematic review on the concepts of terminology [22]. Interests on decisions about life-sustaining treatments gradually increased worldwide from the 1990s, as evident by the Patient Self-determination Act of 1990 in the United States [23], Palliative and Hospice Care Act of 2000 in Taiwan, Mental Capacity Act of 2005 in the United Kingdom, Guidelines for End-of-Life Care Decision-Making Process of 2007 in Japan, and Act on Decisions on Life-Sustaining Treatment for Patients in Hospice and Palliative Care or at the End-of-Life of 2016 in Korea [24]. The yearly trends of terms also seem to have changed with advances in policies. The use of the word “intensive” drastically rose after 2015, suggesting that there were active studies on critical care medicine involving “intensive care unit” or “intensive care.” The use of the words “nurse” and “cancer,” which are often used in collocation with critical care medicine, also increased recently. ICU nurses are healthcare providers who observe EOL patients the most closely and share the most experiences with them and their families [25]. They share the final days of patients’ lives with them and their families and play an essential role in providing EOL care [26]. Additionally, the use of words “home” and “family” also increased, which shows that various forms of services are provided beyond the hospital, such as at home and other facilities, and that EOL care should include not only patients, but also their families. In fact, various forms of hospice services are provided, such as inpatient and home hospice services. The use of the word “dementia” increased most recently. The World Health Organization has expanded the target of palliative care to chronic diseases such as cancer, AIDS, cardiovascular disease, kidney failure, chronic respiratory disease, diabetes mellitus, multiple sclerosis, Parkinson’s disease, dementia, rheumatoid arthritis, and chronic liver disease [27]. The word “child” was relatively less frequent, which suggests that although about 1.2 million children are estimated to require pediatric palliative care worldwide [28] and separate pediatric palliative care systems are being established [29], relevant research is deficient. Additionally, although not derived from the research results, topics related to spirituality, religious beliefs, and emotional wellbeing to manage depression and quality of life in end-stage patients [30,31,32] have been studied recently.

Third, the LDA algorithm identified eight major research topics. It indicated that many studies have been conducted on the ethical problems associated with EOL, such as treatment withdrawal, treatment withholding, and euthanasia. Advances in the life-sustaining treatment system in countries like the United States, United Kingdom, and South Korea were sparked from lawsuits for ethical decisions, such as in the cases of Karen Ann Quinlan [33], Nancy Beth Cruzan [34], Terri Schiavo [35], and Grandmother Kim [36]. Consequently, many ethical and legal studies on the withdrawal and withholding of treatment were conducted; however, a concrete solution is still desired. Studies have been conducted on the quality of life of EOL patients, such as what they prefer and how their symptoms are examined [37]. Some studies suggested that when providing EOL care, caregivers also need to receive care, since they share a similar perception with the patient [38], and spiritual support is helpful for patients [39]. Some studies also highlighted the need for education programs regarding EOL care for healthcare providers, patients, families, and students; additionally, some of these studies examined their communication experiences. Several studies identified and described EOL communication skills for physicians and nurses [40,41]. According to some studies, most patients wanted as much information about their condition as possible, regardless of whether they were good or bad news [42]. Additional studies are needed to examine how healthcare providers could communicate with patients in various ethical situations they might face. Particularly, owing to the nature of the disease, many studies on life-sustaining treatments were conducted on older adults. Similar to the word frequency and yearly trends, there were studies on both inpatient and home hospice care.

The concept of EOL is being established internationally. However, it is still in its infancy, and studies on withholding, withdrawal, and writing forms such as advance directives and Physician orders for life-sustaining treatment are dominating from the point of view of physicians. A nurse’s role in providing EOL care is also diverse, such as the various aspects of the word “care.” There are research topics that need to be continuously developed to include the kind of care to be provided or how to provide care and identify the patient’s preference for self-determination. Moreover, in order to provide effective and evidence-based palliative care nursing, education plans for healthcare providers, patients, and caregivers, as well as studies on patients with children, dementia, and psychological illness are needed. These can also be linked to new knowledge generation in EOL care in nursing research. As EOL care in nursing has expanded from hospital- to home-based, it is suggested that both clinical nurses and home-based nurses should participate in the life-sustaining treatment process.

### 4.1. Strengths and Limitations

Text network analysis is a document analysis method, and unlike systematic literature review, a much larger number of articles (7419) could be analyzed.

In addition, unlike the method of classifying the research contents within the framework of the subject classification already known through a deductive approach for a limited amount of articles, it was possible to grasp the important semantic context of the study through centrality, semantic network, and contiguous sequence word analysis with an inductive approach based on objective data, excluding the subjectivity of the researcher in this study.

After that, statistical keywords were combined to analyze LDA topics to derive and interpret meaningful topics through the discussion of experts. However, the process of checking the context in which keywords are used in a large amount of text data and interpreting the results required a lot of time and effort. In order to understand the statistical analysis results, we tried to increase the validity of the results through the process of repeatedly reading the abstract, which is the original data, and checking the sentences in which the keywords were used.

This study is significant as it analyzed a large volume of existing literature and examined the main research trends in EOL care and nursing, based on core keywords and meaning structures. It was intended to collect literature on the subject of “end-of-life care” using “nursing” as the domain. Since it was difficult to limit nursing to domain, search word was used by combining subject term and domain for the title and abstract of the literature in the related databases. The words “end-of-life care” and “nursing” were naturally included in many cases, so they were evaluated highly in frequency analysis. However, this was considered because the main purpose was to check whether there are other related words, and to determine which research topics were derived by grasping the association between words. Furthermore, it is a mixed-methods study that not only examined the quantitative aspects of word frequency and centrality but also analyzed the qualitative aspects using the LDA method. Through these analyses, we determined the research trends in life-sustaining treatments in the field of nursing and identified areas that are inadequately studied. However, we excluded all studies that were published in languages other than English. Particularly, there were some similar studies published in Korean [43] that had poor readability and relatively less relevant literature. Moreover, these studies may have introduced errors in understanding the overall research trend.

### 4.2. Implications for Nursing and Health Policy

Research on various relevant topics is essential to provide quality EOL care and nursing. It was significant that the text network analysis was performed in a multidisciplinary way in nursing unlike the existing literature review method. Based on the findings of this study, we highlight the need for further studies on pediatric palliative care as well as communication and education programs for nursing students, patients, caregivers, and nurses. Moreover, it is necessary to encourage research on instructing decisions regarding life-sustaining treatments for patients with noncancer chronic diseases and healthy individuals, in addition to providing care for critically ill patients. Furthermore, the findings of this study can be useful for devising mid- to long-term research strategies on palliative and end-of-life care by nurses.

## 5. Conclusions

This study examined the major trends of research on EOL care and nursing, by year, based on frequency and centrality, by performing a text network analysis on all studies published in English until 2019. The results showed that keywords in EOL care and nursing research changed based on the trends of the time. We identified eight subtopics by combining meaningful keywords extracted using the LDA algorithm. Combining the frequently used words showed that previous studies had examined the place of EOL care, the nurses’ roles, and relevant documents. Furthermore, research on life-sustaining treatments was mostly conducted on cancer patients and critically ill patients, with relative disregard to the palliative care for children, communication, and education.

## Figures and Tables

**Figure 1 ijerph-18-00313-f001:**
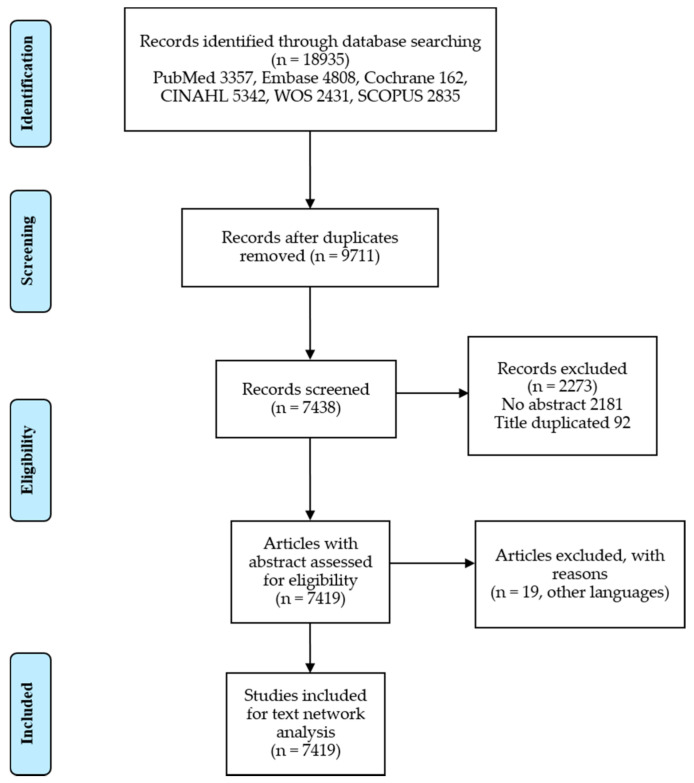
Process of article selection.

**Figure 2 ijerph-18-00313-f002:**
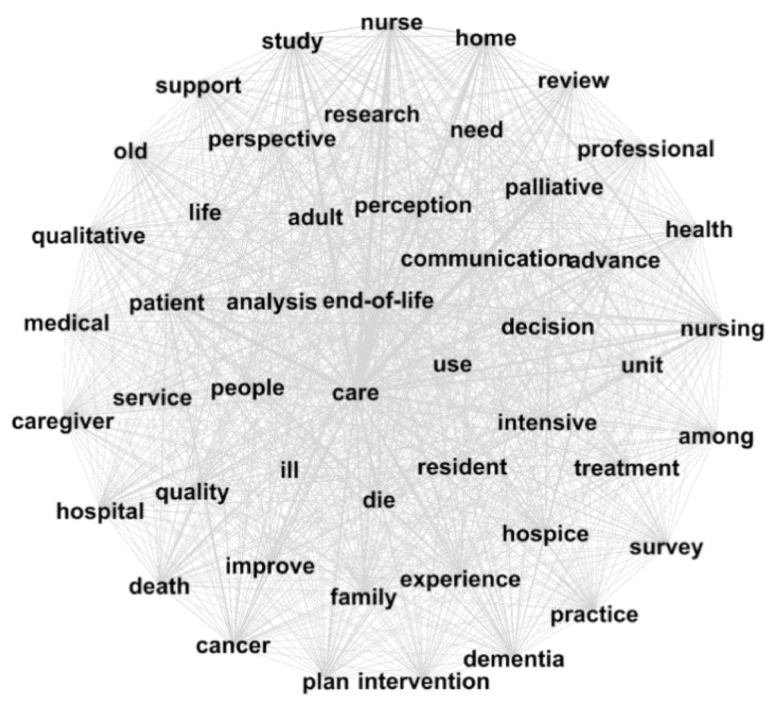
Semantic network analysis of EOL.

**Figure 3 ijerph-18-00313-f003:**
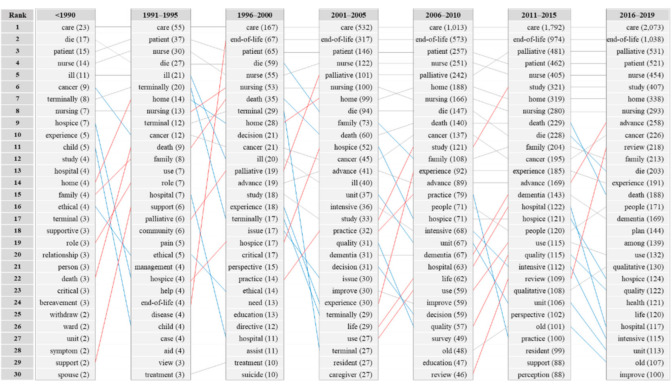
Keywords by five-years cycle.

**Table 1 ijerph-18-00313-t001:** Search criteria and process.

Source	Query		Result
PubMed	#1	“End-of-life care” [All Fields] AND (“nursing” [Subheading] OR “nursing” [All Fields] OR “nursing” [MeSH Terms])	3357
Embase	#1	end-of-life care’ AND (‘nursing’/de OR ‘nursing’ OR ‘nursing service’ OR ‘nursing service, hospital’ OR ‘nursing services’ OR ‘nursing support’ OR ‘nursing, private duty’ OR ‘nursing, supervisory’ OR ‘office nursing’ OR ‘private duty nursing’)	4808
Cochrane	#1	End-of-life care	553
	#2	MeSH descriptor: [Nursing] explode all trees	3192
	#3	nursing	31,911
	#4	#2 OR #3	32,125
	#5	#1 AND #4	162
CINAHL (CINAHLHeadings)	#1	“End-of-life care”	16,884
	#2	nursing	657,389
	#3	#1 AND #2	5342
Web of Science	#1	“End-of-life care” AND nursing	2431
Scopus	#1	“End-of-life care” AND nursing	2835

**Table 2 ijerph-18-00313-t002:** Top 30 keywords that emerged from the end-of-life (EOL) care studies.

Rank	Keyword	F	Keyword	Degree Centrality	Keyword	Betweenness Centrality
1	care	5655	care	0.6991	care	0.2583
2	end-of-life	2973	end-of-life	0.5304	end-of-life	0.1253
3	patient	1503	patient	0.3948	patient	0.0620
4	palliative	1380	palliative	0.3647	palliative	0.0450
5	nurse	1329	nurse	0.3391	nurse	0.0404
6	home	990	study	0.3072	study	0.0319
7	nursing	914	home	0.2833	nursing	0.0308
8	study	907	nursing	0.2833	die	0.0308
9	die	775	die	0.2700	death	0.0258
10	death	664	cancer	0.2448	home	0.0246
11	cancer	645	death	0.2361	cancer	0.0170
12	family	619	advance	0.2079	hospice	0.0118
13	advance	579	family	0.2063	family	0.0112
14	experience	524	experience	0.1971	experience	0.0112
15	dementia	414	use	0.1867	advance	0.0107
16	hospice	396	hospice	0.1655	use	0.0097
17	review	393	hospital	0.1649	hospital	0.0076
18	people	388	people	0.1606	qualitative	0.0072
19	use	348	life	0.1568	life	0.0070
20	hospital	348	among	0.1548	health	0.0068
21	intensive	339	review	0.1541	review	0.0068
22	quality	335	health	0.1530	practice	0.0060
23	unit	334	practice	0.1521	people	0.0059
24	practice	322	quality	0.1515	quality	0.0053
25	life	287	dementia	0.1513	dementia	0.0052
26	old	282	qualitative	0.1437	perspective	0.0052
27	health	281	unit	0.1434	among	0.0050
28	among	280	old	0.1363	support	0.0047
29	qualitative	276	survey	0.1350	old	0.0043
30	resident	274	intensive	0.1346	research	0.0042

EOL: end-of-life, F: frequency.

**Table 3 ijerph-18-00313-t003:** Frequency for the top 20 contiguous sequence keywords according to the number of adjacent words from EOL studies.

Rank	Keywords (n = 2)		F	Keyword (n = 3)			F
1	end-of-life	care	1829	intensive	care	unit	255
2	palliative	care	1071	advance	care	plan	197
3	nursing	home	390	nursing	home	resident	127
4	intensive	care	327	palliative	end-of-life	care	116
5	care	unit	293	care	die	patient	83
6	advance	care	234	improve	end-of-life	care	81
7	care	plan	227	quality	end-of-life	care	73
8	cancer	patient	214	care	intensive	care	61
9	care	nurse	194	provide	end-of-life	care	60
10	care	die	194	care	nursing	home	60
11	qualitative	study	176	ill	cancer	patient	59
12	care	home	171	terminally	ill	cancer	56
13	terminally	ill	167	critical	care	nurse	54
14	home	resident	160	terminally	ill	patient	52
15	care	patient	153	end-of-life	care	intensive	49
16	die	patient	149	end-of-life	care	patient	48
17	critical	care	141	end-of-life	care	people	47
18	nurse	home	140	end-of-life	care	nursing	42
19	systematic	review	137	palliative	care	patient	41
20	palliative	end-of-life	133	long-term	care	facility	40
	long-term	care	133				

EOL: end-of-life, F: frequency.

**Table 4 ijerph-18-00313-t004:** Abstract topic analysis.

Subtopic Groups	Keywords (Weight)
1. Ethical problems of the decision to end the treatment	Decision (0.025), ethical (0.017), euthanasia (0.016), sedation (0.015), withdrawal (0.012), make (0.011), case (0.010), ethic (0.010), medical (0.009), practice (0.009), treatment (0.009), decision-making (0.008), suicide (0.007), process (0.007), request (0.007), patient (0.006), statement (0.006), physician (0.006), nurse (0.006), withhold (0.006)
2. Symptom management to improve the quality of life	Patient (0.044), care (0.037), palliative (0.023), cancer (0.021), review (0.014), intervention (0.012), disease (0.011), symptom (0.010), quality (0.009), end-of-life (0.009), study (0.009), use (0.008), management (0.008), include (0.008), treatment (0.007), need (0.007), research (0.006), pain (0.006), life (0.006), identify (0.006)
3. Development of EOL knowledge education programs	Care (0.060), palliative (0.021), end-of-life (0.021), nurse (0.021), nursing (0.015), education (0.013), student (0.010), practice (0.010), program (0.009), provide (0.009), knowledge (0.009), staff (0.007), improve (0.007), eol (0.007), use (0.007), need (0.006), research (0.006), include (0.006), educational (0.005), develop (0.005)
4. Advanced care planning for older adults	Advance (0.036), dementia (0.031), preference (0.018), treatment (0.016), end-of-life (0.015), plan (0.015), patient (0.015), decision (0.015), ACP (0.014), directive (0.013), discussion (0.011), old (0.010), AD (0.008), make (0.007), wish (0.007), care (0.006), resident (0.006), physician (0.006), adult (0.006), people (0.006)
5. Home-based hospice	Care (0.057), home (0.036), people (0.016), death (0.016), die (0.016), service (0.015), palliative (0.013), hospital (0.013), end-of-life (0.013), study (0.009), need (0.009), place (0.008), staff (0.008), health (0.008), resident (0.008), patient (0.008), community (0.007), support (0.007), nursing (0.007), carers (0.007)
6. Communication experiences	Care (0.045), patient (0.024), nurse (0.024), family (0.020), end-of-life (0.017), die (0.010), experience (0.010), study (0.010), death (0.008), provide (0.008), use (0.007), need (0.007), professional (0.006), support (0.006), interview (0.006), member (0.005), practice (0.005), communication (0.005), health (0.005), research (0.005)
7. Survey of patient symptoms	Patient (0.026), care (0.017), family (0.014), nurse (0.013), study (0.013), death (0.012), die (0.011), caregiver (0.011), symptom (0.010), pain (0.010), quality (0.010), use (0.009), result (0.008), questionnaire (0.008), cancer (0.008), score (0.008), survey (0.008), end-of-life (0.007), measure (0.007), scale (0.007)
8. Analysis of considering patients’ preferences	Care (0.039), hospice (0.026), patient (0.025), home (0.018), resident (0.015), hospital (0.012), use (0.012), death (0.011), end-of-life (0.010), nursing (0.009), die (0.009), facility (0.008), day (0.008), year (0.007), study (0.007), CI (0.007), EOL (0.007), result (0.007), likely (0.006), receive (0.006)

ACP (advanced care planning), AD (advanced directives), CI (confidence interval).

## Data Availability

The data that support the findings of this study are available from the corresponding author, upon reasonable request.

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
