# Peer review of "A Network Analysis of Research Topics and Trends in End-of-Life Care and Nursing"

_ijerph, 2021, doi:10.3390/ijerph18010313_

Round 1
Reviewer 1 Report
I appreciate the tremendous amount of time that this study required in sourcing articles and preparing the titles and abstracts for the analyses that were undertaken. I did wonder though if the main trends might have been discovered by an analysis solely of the key words which accompany most articles. Such an exercise could have been completed much quicker. The author might wish to comment on the value of key-words or at least on how the usefulness of key words identified by authors could be improved.
I am not familiar with the text analysis procedures used by the authors. The clarity of the description could be improved for novices such as me. In particular I did not follow how the Latent Dirichlet Allocation (LDA) method was applied (line 99) and how it yielded the information contained in Table 4. Did the subgroupings emerge from the analysis or were they identified by the authors? I note that the same keywords appear in different subgroupings so how were the subgrouping distinct from one another. This section seemed to me to contain the most useful insights from the study so further details about the methodology are necessary.
The main limitation of using a word-based approach is that the meaning of the word is not captured. For example 'care' can refer to many diverse aspects. It seems that few adjectives were associated with 'care' that could inform the nature of the care provided (Table 3) such as personal care; loving care; tender care; emotional care; spiritual care. Surely these are the aspects of care that crucial for nurses.
I was also puzzled that the analysis of abstracts used only nouns and adjective (line 82) when verbs that describes actions would also seem to be important.
The changes in word usage over time needs to take into account the increase in the number of papers included in more recent years. As well as the actual numbers of mentions, the percentage of mentions in the papers reviewed in each time period could be given. One small discrepancy, I did not follow how the text in lines 145 to 147 corresponded to the numbers shown in Table 2.
The authors rightly make the point about the relative lack of attention given to children and end-of-life care but there are other omissions worthy of note, such as patients' emotional wellbeing, spirituality, religious beliefs, depression and quality of life; all of which relate to the contribution that nurses can make to end-of-life care.
The authors could usefully reflect critically on the limitations of the approach they used and how further research using these methods could be improved and the new insights they would provide. I am not convinced that the time and effort involved is good use of resources.
Author Response
Response to Reviewer 1 Comments
We appreciate the extensive review and helpful comments of the reviewers. Our manuscript was revised according to these comments, and we consulted a professional, scientific English editing service for English language revision.
Comment 1: I did wonder though if the main trends might have been discovered by an analysis solely of the key words which accompany most articles. Such an exercise could have been completed much quicker. The author might wish to comment on the value of key-words or at least on how the usefulness of key words identified by authors could be improved.
Response 1: Thank you for the opportunity to revise our manuscript. This comment seems to apply to method, result and discussion parts in general. We believe that responses to additional comments will be a response to this comment.
Comment 2: I am not familiar with the text analysis procedures used by the authors. The clarity of the description could be improved for novices such as me. In particular I did not follow how the Latent Dirichlet Allocation (LDA) method was applied (line 99) and how it yielded the information contained in Table 4. Did the subgroupings emerge from the analysis or were they identified by the authors? I note that the same keywords appear in different subgroupings so how were the subgrouping distinct from one another. This section seemed to me to contain the most useful insights from the study so further details about the methodology are necessary.
Response 2: We have added an explanation in section 2.5 to help you understand LDA. In addition, the keyword set in the right column of Table 4 is automatically calculated by LDA, and the research team analyzed it and named it in the left column. It is common in LDA for a word to appear in multiple topics, that is, a word can be used in multiple contexts. Moreover, in section 2.5, we have added additional explanations with examples as follows:
Line 106 onward: LDA finds topics that are commonly covered in several documents. For example, if words such as “dog,” “puppy,” “cute,” “snack,” and “walk” frequently appear together in multiple documents, LDA collects these words and treats them as a topic. A topic is a set of words that are automatically calculated through computation. LDA allocates words by iteratively executing an algorithm to find the best topic. A topic is a set of words and not a name. Therefore, topic analysts need to see [dog, puppy, cute, snack, walk] and name them, such as “pet dog;” therefore, expertise for data interpretation is required. Also, “animal hospital” topics such as [dog, sick, hospital, emergency] may additionally be discovered. In this case, it is possible that one word, such as “dog,” appears in multiple topics.
Comment 3: The main limitation of using a word-based approach is that the meaning of the word is not captured. For example 'care' can refer to many diverse aspects. It seems that few adjectives were associated with 'care' that could inform the nature of the care provided (Table 3) such as personal care; loving care; tender care; emotional care; spiritual care. Surely these are the aspects of care that crucial for nurses.
Response 3: Thank you for highlighting this issue. We agree with your comment. We know that many diverse aspects mentioned by you are important in care by nurses. However, it was difficult to find words extracted from “the two- and three-word processing for the top 20” based on the research method. Diverse aspects of care (personal care, tender care, spiritual care) were added to the discussion with reference to the comment as follows:
Line 244 onward: The most frequent word “care” has many diverse aspects in nursing—spiritual care [20], tender care [21], and personal care [22] for life sustaining patients, but these were not included in the top results extracted during analyses.
Comment 4: I was also puzzled that the analysis of abstracts used only nouns and adjective (line 82) when verbs that describes actions would also seem to be important.
Response 4: In general, which part of speech to use is optional depending on the purpose of analysis. Nouns are used to identify the subject of an article, and adjectives are used together with nouns to identify auxiliary features of the subject. This explanation has been added in section 2.3.
As the reviewer mentioned, verbs can be used to analyze behavior; however, we needed to consider the specificity of the abstract. Abstracts describe key topics in short paragraphs. In this process, the verbs in the abstract tend to appear more intensively in general research-related actions such as “examine,” “describe,” “propose,” and “demonstrate” than specific actions. Therefore, these words were excluded from being dominant in the topic.
Therefore, we added the sentences and references as follows:
Line 84 onward: “In general, which part of speech to use may be selected depending on the purpose of analysis. Nouns are used to identify the subject of an article, and adjectives are used together with nouns to identify auxiliary features of the subject [14,15].”
Comment 5: The changes in word usage over time needs to take into account the increase in the number of papers included in more recent years. As well as the actual numbers of mentions, the percentage of mentions in the papers reviewed in each time period could be given. One small discrepancy, I did not follow how the text in lines 145 to 147 corresponded to the numbers shown in Table 2.
Response 5: Thank you for highlighting this confusing error. We apologize that it was described incorrectly in the process of modifying the table. This part was re-described according to the contents of Table 2 (Lines 162–163).
Comment 6: The authors rightly make the point about the relative lack of attention given to children and end-of-life care but there are other omissions worthy of note, such as patients' emotional wellbeing, spirituality, religious beliefs, depression and quality of life; all of which relate to the contribution that nurses can make to end-of-life care.
Response 6: Thank you for giving specific comments on what is omitted. The concepts given in comments were not derived as specific words in the results of this study. However, we agree that it is a topic of future study for EOL care in nursing; the reference was supplemented and added in the discussion (Lines 272–275).
Reviewer 2 Report
On the whole the article is well structured and clear in the exposition of the methodologies and results.
I would improve the abstract which is not exhaustive enough of the content reported in the article.
Author Response
Response to Reviewer 2 Comments
Comment 1: I would improve the abstract which is not exhaustive enough of the content reported in the article.
Response 1: Thank you for your thoughtful comments and recommendations. An abstract sentence was added to emphasize that eight topics were derived as a result of topic modeling by reflecting the comments (Lines 17–18).
Reviewer 3 Report
Overall, the paper is addressing a significant industry; aiming to find trends for End-of-life care. However, some major revisions need to be done before the publication of this paper.
Research methodology needs to be improved.
Please elaborate on what were the search criteria for choosing the most relevant keywords. Please read the related literature, whether there is a systematic literature review to adopt its keywords accordingly.
How did the authors manage to remove the duplicate publications? Please review the below paper and follow the process of data collection accordingly:
Dehghani, M., & Kim, K. J. (2019). Past and present research on wearable technologies: bibliometric and cluster analyses of published research from 2000 to 2016. International Journal of Innovation and Technology Management, 16(01), 1950007.
In the results section, the current paper seems more like a report. The results are written in a superficial approach. Section 3.1 and 3.5 can be improved. For example, the authors can interpret the themes and provide current status and future direction. Please read the below papers:
Dehghani, M., Mashatan, A., & Kennedy, R. W. (2020). Innovation within networks–patent strategies for blockchain technology. Journal of Business & Industrial Marketing.
Gu, D., Li, J., Li, X., & Liang, C. (2017). Visualizing the knowledge structure and evolution of big data research in healthcare informatics. International journal of medical informatics, 98, 22-32.
Discussion section must be improved. Please add a new section including some robust healthcare implications.
Author Response
Response to Reviewer 3 Comments
Comment 1: Research methodology needs to be improved. Please elaborate on what were the search criteria for choosing the most relevant keywords. Please read the related literature, whether there is a systematic literature review to adopt its keywords accordingly.
How did the authors manage to remove the duplicate publications? Please review the below paper and follow the process of data collection accordingly:
Dehghani, M., & Kim, K. J. (2019). Past and present research on wearable technologies: bibliometric and cluster analyses of published research from 2000 to 2016. International Journal of Innovation and Technology Management, 16(01), 1950007.
Response 1: We added the process regarding article selection in Figure 2 and how it modified the PRISMA flow diagram. Further, we managed the duplicated publication in 2 phases. We used Endnote program in first phase and Excel sorting function in the second phase. Moreover, we added the process in line 101 for clarification about the data collection procedure.
Figure 1. Process of article selection.
Comment 2: In the results section, the current paper seems more like a report. The results are written in a superficial approach. Section 3.1 and 3.5 can be improved. For example, the authors can interpret the themes and provide current status and future direction. Please read the below papers:
Dehghani, M., Mashatan, A., & Kennedy, R. W. (2020). Innovation within networks–patent strategies for blockchain technology. Journal of Business & Industrial Marketing.
Gu, D., Li, J., Li, X., & Liang, C. (2017). Visualizing the knowledge structure and evolution of big data research in healthcare informatics. International journal of medical informatics, 98, 22-32.
Response 2: Thank you for your thoughtful comments and recommendations. The recommended papers were very helpful in revising the results. These have been added to reflect the current state and provide future directions by redescribing the results in sections 3.1 and 3.5. (Lines 134–136, 138, 140–141, 143–145, 195–198, 200-202, 205–207, 211–213, 214-216, 218–220, 222–224, 227–228).
Comment 3: Discussion section must be improved. Please add a new section including some robust healthcare implications.
Response 3: We agree with the reviewer’s comments, and to reflect the same, a paragraph has been added to the part of the discussion, as follows.
Line 296 onward: The concept of EOL is being established internationally. However, it is still in its infancy, and studies on withholding, withdrawal, and writing forms such as advance directives and POLST are dominating from the point of view of physicians. A nurse’s role in providing EOL care is also diverse, such as the various aspects of the word “care.” There are research topics that need to be continuously developed to include the kind of care to be provided or how to provide care and identify the patient's preference for self-determination. Moreover, in order to provide effective and evidence-based palliative care nursing, education plans for healthcare providers, patients, and caregivers, as well as studies on patients with children, dementia, and psychological illness are needed. These can also be linked to new knowledge generation in EOL care in nursing research. As EOL care in nursing has expanded from hospital- to home-based, it is suggested that both clinical nurses and home-based nurses should participate in the life-sustaining treatment process.

Round 2
Reviewer 1 Report
I appreciate the response of the authors to my comments but they failed to address my final comment that related to the limitations of this methodology. The literature search they undertook used certain key-words in this instance: 'end-of-life care' and 'nursing'. It is no surprise then that these feature as the most common words as the outcomes were pre-determined. The analysis of abstracts added additional words but as the authors acknowledge the subtleties of meaning associated with the words or phrases is not captured. A full reading of the article is crucial to a fuller understanding of the issues which a systematic review of pertinent articles could elucidate. My concern is that readers need to appreciate the limitations to the approach adopted by the researchers so that they can understand the role of other forms of literature reviews and which are likely to be the more insightful.
Author Response
Response to Reviewer 1 Comments
Comment 1: I appreciate the response of the authors to my comments but they failed to address my final comment that related to the limitations of this methodology.
Response 1: Thank you for granting us the opportunity to revise manuscript again. We do not want to disappoint you again, and we did our best to avoid repeating the same mistakes.
Readers may have difficulty understanding it, since the text network analysis is different from the existing literature review method. Based on your valuable comments, we have added more detailed descriptions and limitations of the research method in the part of method, results, discussion.
Comment 2: The literature search they undertook used certain key-words in this instance: 'end-of-life care' and 'nursing'. It is no surprise then that these feature as the most common words as the outcomes were pre-determined.
Response 2: We agree with the reviewer’s assessment of the methodology. Searching keywords may have been affected in selecting the most common words. However, the goal is to collect literature on end-of-life care topics in the nursing domain. As it is difficult to limit only the literature corresponding to the nursing domain, a search word was used by combining the subject term and domain for the title and abstract of the literature in the related database.
In terms of the literature extraction process, you can see more detail through the newly added Table1.
Table 1. Search criteria and process.
|
Source |
Query |
Result |
|
|
Pubmed |
#1 |
"End-of-life care"[All Fields] AND ("nursing"[Subheading] OR "nursing"[All Fields] OR "nursing"[MeSH Terms]) |
3,357 |
|
EMBASE |
#1 |
end-of-life care' AND ('nursing'/de OR 'nursing' OR 'nursing service' OR 'nursing service, hospital' OR 'nursing services' OR 'nursing support' OR 'nursing, private duty' OR 'nursing, supervisory' OR 'office nursing' OR 'private duty nursing') |
4,808 |
|
Cochrane |
#1 |
End-of-life care |
553 |
|
#2 |
MeSH descriptor: [Nursing] explode all trees |
3,192 |
|
|
#3 |
nursing |
31,911 |
|
|
#4 |
#2 OR #3 |
32,125 |
|
|
#5 |
#1 AND #4 |
162 |
|
|
CINAHL |
#1 |
"End-of-life care" |
16,884 |
|
#2 |
nursing |
657,389 |
|
|
#3 |
#1 AND #2 |
5,342 |
|
|
Web of Science |
#1 |
"End-of-life care" AND nursing |
2,431 |
|
Scopus |
#1 |
"End-of-life care" AND nursing |
2,835 |
In the analysis results, the number of analyzes is highly evaluated because it includes a lot of words directly used in the search: end-of-life care and nursing. Even considering this reason, the main purpose of the analysis process is to determine what other words are related to, how they relate, and what topics are derived.
Line 371 onward:
|
4.1. Strengths and Limitations Text network analysis is a document analysis method, and unlike systematic literature review, a much larger number of articles (7,419) could be analyzed. In addition, unlike the method of classifying the research contents within the framework of the subject classification already known through a deductive approach for a limited amount of articles, it was possible to grasp the important semantic context of the study through centrality, semantic network and contiguous sequence word analysis with an inductive approach based on objective data, excluding the subjectivity of the researcher in this study. After that, statistical keywords were combined to analyze LDA topics to derive and interpret meaningful topics through the discussion of experts. However the process of checking the context in which keywords are used in a large amount of text data and interpreting the results required a lot of time and effort. In order to understand the statistical analysis results, we tried to increase the validity of the results through the process of repeatedly reading the abstract, which is the original data, and checking the sentences in which the keywords were used. This study is significant as it analyzed a large volume of existing literature and examined the main research trends in EOL care and nursing, based on core keywords and meaning structures. It was intended to collect literature on the subject of “end-of-life care” using "nursing" as the domain. Since it was difficult to limit nursing to domain, search word was used by combining subject term and domain for the title and abstract of the literature in the related databases. The words “end-of-life care” and “nursing” were naturally included in many cases, so they were evaluated highly in frequency analysis. However, this was considered because the main purpose was to check whether there are other related words, and to determine which research topics were derived by grasping the association between words. Furthermore, it is a mixed-methods study that not only examined the quantitative aspects of word frequency and centrality but also analyzed the qualitative aspects using the LDA method. Through these analyses, we determined the research trends in life-sustaining treatments in the field of nursing and identified areas that are inadequately studied. However, we excluded all studies that were published in languages other than English. Particularly, there were some similar studies published in Korean [43] that had poor readability and relatively less relevant literature. Moreover, these studies may have introduced errors in understanding the overall research trend. |
Comment 3: The analysis of abstracts added additional words but as the authors acknowledge the subtleties of meaning associated with the words or phrases is not captured.
Response 3: Thank you for pointing this out. In section 3.1 of result, it described based on the frequency and centrality of words. In section 3.4, it showed phrases which are combination of the words. Phase is a combination of co-occurrence words. The content showed in section 3.4 focuses more on contiguous sequences as well as co-occurrence. It showed not only words that appear together, but also in what order they appear. Co-occurrence only shows word-to-word connectivity, but n-gram (n consecutive words) also serves to show a directed path between words.
Line 122 onward:
|
3.1. Core Keywords that Emerged from the EOL Care Studies Keyword ranking was based on the frequency of the appearance of keywords used in the EOL studies’ titles (i.e., the number of articles with keywords, degree centrality, and betweenness centrality). Table 2 lists the top 30 core keywords by criteria. The frequency order of the simple appearance was as follows: “care” (5,655), “end-of-life” (2,973), “patient” (1,503), “palliative” (1,380), and “nurse” (1,329). Identical results were observed in the top five degree and betweenness centralities. The top five ranked words for the degree and betweenness centralities were as follows: “care” (0.6991/0.2583), “end-of-life” (0.5304/0.1253), “patient” (0.3948/0.0620), “palliative” (0.3647/0.0450), and “nurse” (0.3391/0.0404). Degree centrality is to find out which keywords have a lot of connection to the research topic. The words that are ranked at the top are the ones that are being treated in focus. On the other hand, it can be seen that research is insufficient for words such as “dementia” (rank 25th), “qualitative” (rank 26th), and “old” (rank 28th) that are ranked in the lower part. Betweenness centrality is a measure of the degree to which one keyword acts as an intermediary to another. Although degree centrality and betweenness centrality show similar patterns, there are keywords that show differences. Keywords with high betweenness centrality can have a great influence on controlling the flow of information within the network. The word “support”, ranked 28th in betweenness centrality, does not exist in the degree centrality ranking. This means that "support" has a greater role in affecting other words than its central role in words. The frequency, and degree and betweenness centralities for the top eight words were similar. However, in terms of frequency and degree centrality, the word “home” ranked 6th and 7th, respectively, and its betweenness centrality ranked lower at 10th. The word “home” is more connected to the other frequently-used words seen as high in importance. However, it can be interpreted that centrality was relatively low among other words. The word “dementia” ranked 15th for frequency in the titles, and both degree and betweenness centralities were lower at the 25th rank. Thus, although, “dementia” is often mentioned, its importance and centrality are relatively low. This difference in analysis shows that the frequency of the keyword is not necessarily proportional to the size of centrality. In other words, it means that even keywords with high frequency may not be relatively centered. Compared to “family” and “people,” “patient” appeared more frequently in the EOL care studies and showed a higher connection and centrality, thus indicating that more studies were conducted on patients in hospitals rather than on general and healthy population. Furthermore, words with similar forms such as “nurse” and “nursing,” “die” and “death,” and “quality” and “qualitative” were close to each other and had similar rankings in frequency and centrality. Since words of the same meaning with different forms are considered to have similar positioning, priority results are reliable. |
Line 195 onward:
|
3.4. Frequency According to Contiguous Sequence Word Analysis of Research Titles Tables 3 shows the results of contiguous sequence word analysis. Owing to low frequencies, the cases of four- and five-word expressions of contiguous words were excluded; the two- and three-word processing for the top 20 were shown as follows. Contiguous sequence is not a simple frequency of two and three word phases. This section shows not only the co-occurrence of words appearing together based on the centrality between words, but also in what order they appear by focusing on the contiguous sequence. This represents a directed path between words. |
Comment 4: A full reading of the article is crucial to a fuller understanding of the issues which a systematic review of pertinent articles could elucidate.
Response 4: Thank you for pointing this out. Text network analysis differs in systematic literature review and method of understanding and interpreting the entire content. The text network analysis used in this study is to derive meaning through combinations of unstructured words.
It is not collecting the results of each study, but predicting what research topics were studied using the words used. In this study, we studied at the research status using such a text network analysis method. However, as you pointed out, it seems necessary to check what differences and similarities they have compared to the systematic literature review methods. This part was included in the discussion.
Line 371 onward:
|
4.1. Strengths and Limitations Text network analysis is a document analysis method, and unlike systematic literature review, a much larger number of articles (7,419) could be analyzed. In addition, unlike the method of classifying the research contents within the framework of the subject classification already known through a deductive approach for a limited amount of articles, it was possible to grasp the important semantic context of the study through centrality, semantic network and contiguous sequence word analysis with an inductive approach based on objective data, excluding the subjectivity of the researcher in this study. After that, statistical keywords were combined to analyze LDA topics to derive and interpret meaningful topics through the discussion of experts. However the process of checking the context in which keywords are used in a large amount of text data and interpreting the results required a lot of time and effort. In order to understand the statistical analysis results, we tried to increase the validity of the results through the process of repeatedly reading the abstract, which is the original data, and checking the sentences in which the keywords were used. This study is significant as it analyzed a large volume of existing literature and examined the main research trends in EOL care and nursing, based on core keywords and meaning structures. It was intended to collect literature on the subject of “end-of-life care” using "nursing" as the domain. Since it was difficult to limit nursing to domain, search word was used by combining subject term and domain for the title and abstract of the literature in the related databases. The words “end-of-life care” and “nursing” were naturally included in many cases, so they were evaluated highly in frequency analysis. However, this was considered because the main purpose was to check whether there are other related words, and to determine which research topics were derived by grasping the association between words. Furthermore, it is a mixed-methods study that not only examined the quantitative aspects of word frequency and centrality but also analyzed the qualitative aspects using the LDA method. Through these analyses, we determined the research trends in life-sustaining treatments in the field of nursing and identified areas that are inadequately studied. However, we excluded all studies that were published in languages other than English. Particularly, there were some similar studies published in Korean [44] that had poor readability and relatively less relevant literature. Moreover, these studies may have introduced errors in understanding the overall research trend. |
Comment 5: My concern is that readers need to appreciate the limitations to the approach adopted by the researchers so that they can understand the role of other forms of literature reviews and which are likely to be the more insightful.
Response 5: We agree with the reviewer’s assessment of the method. We aim to grasp the research topic from a macroscopic perspective by analyzing phenomena commonly observed in vast literature. Text network analysis is a research method that quantitatively observes various characteristics of keywords by forming a network using words and their relationships and introducing a well-established network analysis method.
In contrast, LDA topic analysis identifies topics commonly contained in literature based on unsupervised learning. The subject at this time is formed by a combination of keywords based on probability/statistics. Therefore, it is necessary for experts in the field to judge what kind of research content specifically means the combination of keywords (topics) discovered by the statistic program, and the main contribution is to perform meaningful naming.
Line 195 onward:
|
3.4. Frequency According to Contiguous Sequence Word Analysis of Research Titles Tables 3 shows the results of contiguous sequence word analysis. Owing to low frequencies, the cases of four- and five-word expressions of contiguous words were excluded; the two- and three-word processing for the top 20 were shown as follows. Contiguous sequence is not a simple frequency of two and three word phases. This section shows not only the co-occurrence of words appearing together based on the centrality between words, but also in what order they appear by focusing on the contiguous sequence. This represents a directed path between words. |
Line 214 onward:
|
When “patient” and “people” are placed after the words, they represent subjects for nursing performance: “cancer + patient,” “care + patient,” “die + patient,” “care + die + patient,” “ill + cancer + patient,” “terminally + ill + patient,” “end-of-life + care + patient,” “end-of-life + care + people” and “palliative + care + patient”. |
Line 223 onward:
|
LDA topic analysis identifies topics commonly contained in literature based on unsupervised learning. The topics are formed as keywords combination based on statistics. Experts in the relevant field judge the meaning of keyword combinations according to statistics and derive meaningful topics. Seven rounds of LDA were performed with varying number of topics (K = 2, 3, 4, 5, 8, 10, 20). In the case of K=2 and 3, it was difficult to derive meaningful content because the number of subtopics was too small. Since K=20 has a large number of subtopics, there were overlapping topics. Subtopics were grouped through discussion among researchers, and K=8 topics with no overlapping meaning between groups were finally selected (Table 4). |
Reviewer 3 Report
In my previous review, I provided a major revision for the submitted manuscript. However, the authors returned back the manuscript after 4-5 days (that speaks the quality of revision by itself) without including my requested revisions. Only one part of my revisions from the duplicate approach is revised. Therefore, I can not accept the current version of the submitted manuscript. First and foremost, I urge the authors to emphasis on improving the paper from the orientation of results as a whole. As I said earlier, conducting a simple semantic analysis and highlighting most frequent words does not add a contribution to the existing literature. Please read my previous comments and apply them in details. For the new version, I would like to see a figure that explains the five-years cycle of EOL year by year. Also, I would like to see a clear interpretation of subtopic groups (table 4) one by one.
Author Response
Response to Reviewer 3 Comments
Comment 1: In my previous review, I provided a major revision for the submitted manuscript. However, the authors returned back the manuscript after 4-5 days (that speaks the quality of revision by itself) without including my requested revisions. Only one part of my revisions from the duplicate approach is revised. Therefore, I can not accept the current version of the submitted manuscript.
Response 1: Thank you for giving us one more chance to revise. We are very sorry that we have not been sufficiently satisfied with the valuable comments you gave before. It seems that it was a big mistake to rush to finish it as soon as earlier.
We do not want to miss this precious opportunity so we will do our best to revise this manuscript. Thank you for your time to read our responses and revised manuscript again.
Comment 2: First and foremost, I urge the authors to emphasis on improving the paper from the orientation of results as a whole.
Response 2: We thank the reviewer for detailed consideration of the result. We reviewed the two articles you recommended in previous comments repetitively. It was very helpful as experts in the field of research methods organized the research results in detail. We tried to satisfy the comments by inserting additional table and figure and modifying the result interpretation process as a whole even though within a limited revision period to contain the better results. We will describe these amendments in detail in the following comments.
Comment 3: As I said earlier, conducting a simple semantic analysis and highlighting most frequent words does not add a contribution to the existing literature.
Response 3: Thank you for pointing this out. We aim to grasp the research topic from a macroscopic perspective by analyzing phenomena commonly observed in vast literature. Text network analysis is a research method that quantitatively observes various characteristics of keywords by forming a network using words and their relationships, and introducing a well-established network analysis method. Also, the content covered in section 3.4 focuses on contiguous sequences, not simple frequencies. It shows not only words that appear together, but also in what order they appear. Co-occurrence only shows word-to-word connectivity, while n-grams (n consecutive words) also serve to show a directed path between words. In contrast, the LDA topic analysis introduced in the abstract analysis is based on unsupervised learning to identify topics commonly contained in literature. The subject at this time is formed by a combination of keywords based on probability/statistics. Therefore, it is necessary for experts in the field to judge what kind of research content specifically means the combination of keywords (topics) discovered by the statistic program, and the main contribution is to perform meaningful naming. We have added a description of it to the text by referring to the comments.
Line 195 onward:
|
3.4. Frequency According to Contiguous Sequence Word Analysis of Research Titles Tables 3 shows the results of contiguous sequence word analysis. Owing to low frequencies, the cases of four- and five-word expressions of contiguous words were excluded; the two- and three-word processing for the top 20 were shown as follows. Contiguous sequence is not a simple frequency of two and three word phases. This section shows not only the co-occurrence of words appearing together based on the centrality between words, but also in what order they appear by focusing on the contiguous sequence. This represents a directed path between words. |
Line 214 onward:
|
When “patient” and “people” are placed after the words, they represent subjects for nursing performance: “cancer + patient,” “care + patient,” “die + patient,” “care + die + patient,” “ill + cancer + patient,” “terminally + ill + patient,” “end-of-life + care + patient,” “end-of-life + care + people” and “palliative + care + patient”. |
Line 223 onward:
|
LDA topic analysis identifies topics commonly contained in literature based on unsupervised learning. The topics are formed as keywords combination based on statistics. Experts in the relevant field judge the meaning of keyword combinations according to statistics and derive meaningful topics. Seven rounds of LDA were performed with varying number of topics (K = 2, 3, 4, 5, 8, 10, 20). In the case of K=2 and 3, it was difficult to derive meaningful content because the number of subtopics was too small. Since K=20 has a large number of subtopics, there were overlapping topics. Subtopics were grouped through discussion among researchers, and K=8 topics with no overlapping meaning between groups were finally selected (Table 4). |
Comment 4: Please read my previous comments and apply them in details.
Response 4: We will response by carefully reviewing the valuable comments you made last time, sentence by sentence.
Previous comment (1) Research methodology needs to be improved. Please elaborate on what were the search criteria for choosing the most relevant keywords.
Response 4(1): In response to your comments, we've added Table 1 to further check our search criteria.
Table 1. Search criteria and process.
|
Source |
Query |
Result |
|
|
Pubmed |
#1 |
"End-of-life care"[All Fields] AND ("nursing"[Subheading] OR "nursing"[All Fields] OR "nursing"[MeSH Terms]) |
3,357 |
|
EMBASE |
#1 |
end-of-life care' AND ('nursing'/de OR 'nursing' OR 'nursing service' OR 'nursing service, hospital' OR 'nursing services' OR 'nursing support' OR 'nursing, private duty' OR 'nursing, supervisory' OR 'office nursing' OR 'private duty nursing') |
4,808 |
|
Cochrane |
#1 |
End-of-life care |
553 |
|
#2 |
MeSH descriptor: [Nursing] explode all trees |
3,192 |
|
|
#3 |
nursing |
31,911 |
|
|
#4 |
#2 OR #3 |
32,125 |
|
|
#5 |
#1 AND #4 |
162 |
|
|
CINAHL |
#1 |
"End-of-life care" |
16,884 |
|
#2 |
nursing |
657,389 |
|
|
#3 |
#1 AND #2 |
5,342 |
|
|
Web of Science |
#1 |
"End-of-life care" AND nursing |
2,431 |
|
Scopus |
#1 |
"End-of-life care" AND nursing |
2,835 |
Previous comment (2) In the results section, the current paper seems more like a report. The results are written in a superficial approach. Section 3.1 and 3.5 can be improved. For example, the authors can interpret the themes and provide current status and future direction.
Response 4(2): In addition to sections 3.1 and 3.5, a detailed result analysis process has been added for the entire result section.
Line 122 onward:
|
3.1. Core Keywords that Emerged from the EOL Care Studies Keyword ranking was based on the frequency of the appearance of keywords used in the EOL studies’ titles (i.e., the number of articles with keywords, degree centrality, and betweenness centrality). Table 2 lists the top 30 core keywords by criteria. The frequency order of the simple appearance was as follows: “care” (5,655), “end-of-life” (2,973), “patient” (1,503), “palliative” (1,380), and “nurse” (1,329). Identical results were observed in the top five degree and betweenness centralities. The top five ranked words for the degree and betweenness centralities were as follows: “care” (0.6991/0.2583), “end-of-life” (0.5304/0.1253), “patient” (0.3948/0.0620), “palliative” (0.3647/0.0450), and “nurse” (0.3391/0.0404). Degree centrality is to find out which keywords have a lot of connection to the research topic. The words that are ranked at the top are the ones that are being treated in focus. On the other hand, it can be seen that research is insufficient for words such as “dementia” (rank 25th), “qualitative” (rank 26th), and “old” (rank 28th) that are ranked in the lower part. Betweenness centrality is a measure of the degree to which one keyword acts as an intermediary to another. Although degree centrality and betweenness centrality show similar patterns, there are keywords that show differences. Keywords with high betweenness centrality can have a great influence on controlling the flow of information within the network. The word “support”, ranked 28th in betweenness centrality, does not exist in the degree centrality ranking. This means that "support" has a greater role in affecting other words than its central role in words. The frequency, and degree and betweenness centralities for the top eight words were similar. However, in terms of frequency and degree centrality, the word “home” ranked 6th and 7th, respectively, and its betweenness centrality ranked lower at 10th. The word “home” is more connected to the other frequently-used words seen as high in importance. However, it can be interpreted that centrality was relatively low among other words. The word “dementia” ranked 15th for frequency in the titles, and both degree and betweenness centralities were lower at the 25th rank. Thus, although, “dementia” is often mentioned, its importance and centrality are relatively low. This difference in analysis shows that the frequency of the keyword is not necessarily proportional to the size of centrality. In other words, it means that even keywords with high frequency may not be relatively centered. Compared to “family” and “people,” “patient” appeared more frequently in the EOL care studies and showed a higher connection and centrality, thus indicating that more studies were conducted on patients in hospitals rather than on general and healthy population. Furthermore, words with similar forms such as “nurse” and “nursing,” “die” and “death,” and “quality” and “qualitative” were close to each other and had similar rankings in frequency and centrality. Since words of the same meaning with different forms are considered to have similar positioning, priority results are reliable. 3.2. Semantic Network Analysis Figure 2 displays a semantic network diagram that was created to examine the network between core keywords based on degree centrality. These are the nodes ranked in the top 1% with a degree of more than 600. The words in the figure 2 have more than 600 connecting edges. Higher-ranked nodes are located in the center, and lower-ranked nodes are located in the periphery. Through the entire semantic network, it can be seen that words are grouped around “care”. Nodes in the center have a greater influence than nodes located in the periphery. It has a similar ranking to the words located in Table 2. The keyword with the highest degree centrality was “care.” The keywords ranked in the top 1% of degree centrality were “end-of-life,” “patient,” and “palliative,” which were consistent with the most frequently observed keywords. |
Previous comment (3) Discussion section must be improved. Please add a new section including some robust healthcare implications.
Response 4(3): This is already reflected in lines 359 according to the comments you gave last time. Additionally, in section 4.1, we describe the advantages and disadvantages of performing text network analysis that may be unfamiliar to readers.
Line 371 onward:
|
4.1. Strengths and Limitations Text network analysis is a document analysis method, and unlike systematic literature review, a much larger number of articles (7,419) could be analyzed. In addition, unlike the method of classifying the research contents within the framework of the subject classification already known through a deductive approach for a limited amount of articles, it was possible to grasp the important semantic context of the study through centrality, semantic network and contiguous sequence word analysis with an inductive approach based on objective data, excluding the subjectivity of the researcher in this study. After that, statistical keywords were combined to analyze LDA topics to derive and interpret meaningful topics through the discussion of experts. However the process of checking the context in which keywords are used in a large amount of text data and interpreting the results required a lot of time and effort. In order to understand the statistical analysis results, we tried to increase the validity of the results through the process of repeatedly reading the abstract, which is the original data, and checking the sentences in which the keywords were used. This study is significant as it analyzed a large volume of existing literature and examined the main research trends in EOL care and nursing, based on core keywords and meaning structures. It was intended to collect literature on the subject of “end-of-life care” using "nursing" as the domain. Since it was difficult to limit nursing to domain, search word was used by combining subject term and domain for the title and abstract of the literature in the related databases. The words “end-of-life care” and “nursing” were naturally included in many cases, so they were evaluated highly in frequency analysis. However, this was considered because the main purpose was to check whether there are other related words, and to determine which research topics were derived by grasping the association between words. Furthermore, it is a mixed-methods study that not only examined the quantitative aspects of word frequency and centrality but also analyzed the qualitative aspects using the LDA method. Through these analyses, we determined the research trends in life-sustaining treatments in the field of nursing and identified areas that are inadequately studied. However, we excluded all studies that were published in languages other than English. Particularly, there were some similar studies published in Korean [43] that had poor readability and relatively less relevant literature. Moreover, these studies may have introduced errors in understanding the overall research trend. |